# Systems Thinking and Group Concept Mapping for Classification of Marketing Techniques in Mobility Plans

**Stefano Armenia** [1], **Georgios Tsaples** [2,*], **Riccardo Onori** [3], **Alessandro Pompei** [4] and **Robert Magnuszewski** [5]

1 Department of Research, Link Campus University of Rome, 00165 Rome, Italy
2 Department of Business Administration, University of Macedonia, 54636 Thessaloniki, Greece
3 System Dynamics Italian Chapter, SYDIC, 00146 Rome, Italy
4 Department of Mechanical and Aerospace Engineering, Sapienza University of Rome, 00185 Rome, Italy
5 Department of Department of Industrial Engineering Information and Economics, University of L'Aquila, 67100 L'Aquila, Italy
* Correspondence: gtsaples@uom.edu.gr

**Abstract:** Due to the increasing urbanization of the global population, policymakers and academia have shifted their attention toward mobility plans and inquiries on how to best design and implement them. Hence, in order to introduce meaningful and lasting mobility plans, coordination and addressing the needs of a wide number of stakeholders are required. Reaching such a broad target audience may appear challenging for policymakers; nonetheless, the employment of appropriate marketing techniques can make the different stakeholders aware of the problems at stake and inform them about the available solutions. However, the question of which marketing technique to use in order to increase the probability of success for the mobility plan still remains. The purpose of the current paper is to propose a hybrid method that can assist policymakers in categorizing the marketing techniques towards the sustainable urban mobility plan's steps, with the combination of Systems Thinking and modified Group Concept Mapping. The paper concludes with a series of recommendations to policymakers on which marketing technique appears to be the most appropriate under different conditions. The novelties of the paper are the combination of the two methodologies and the practical recommendations that could be useful by policymakers. Moreover, the paper illustrates an example of how to structure and more efficiently use evidence-based policies.

**Keywords:** systems thinking; group concept mapping; marketing techniques; mobility plans; SUMP cycle

## 1. Introduction

The global population is experiencing a period of change and increased urbanization, as in 2018, 74% of European citizens lived in towns and cities and this share is expected to increase in the coming decades [1]. This process of continuous expansion often leads to increase in distances even beyond local municipalities, hence adding new challenges to policymakers. Taking care of adequate functionality of urban areas is, however, of utmost importance. Considering that a significant proportion of European cities already struggle with issues related to clean air and noise pollution, as well as with a high number of road accidents, the improvement and development of urban mobility seems like an unavoidable and urgent need.

Dealing with the new challenges implies that all residents are offered appropriate options and that transportation is safe, with minimal consequence to noise and pollution, with improved efficiency and cost-effectiveness, while contributing to the attractiveness and quality of the urban environment. To this end, policymakers employ "Sustainable Urban Mobility Plans" (SUMP) [2], which serve as a set of guiding principles to define mobility policies that have clear measurable targets and are able to engage all stakeholders.

At the core of each SUMP phase lies the management of the needs and expectations of different stakeholders, hence, in order to introduce meaningful and lasting mobility plans in the appropriate SUMP phase, coordination is of the utmost importance. Reaching such a broad target audience may appear challenging for policymakers; nonetheless, the employment of appropriate marketing techniques can make the different stakeholders aware of the problems at stake and inform them about the available solutions. However, the question of which marketing technique to use in order to increase the probability of success for the mobility plan still remains.

Consequently, the purpose of the current paper is to propose a methodological framework that can assist policymakers in categorizing marketing techniques with steps of Sustainable Urban Mobility Plans. By having more appropriate (or targeted) marketing techniques for each step, policymakers can develop action plans that provide tailored guidelines that can increase the participation of citizens and relevant stakeholders in the formulation of mobility plans.

To achieve the objective of the current paper, a hybrid inductive approach was designed, consisting of systemic (Systems Thinking) as well as mixed multi-criteria (Group Concept mapping) analytic techniques applied to a knowledge base of good practices (GP) that were collected. The practices, either coming from the regions participating in the e-smartec project or regions outside the project, are really valuable for the categorization analysis because they represent concrete outcomes of the implementation of marketing techniques and related methods in a specific context. Thus, through categorization and a deeper investigation of practices and techniques, some insights have been revealed that could be helpful for policymakers to enhance the participatory approach of sustainable mobility plans in their regions.

The structure of the remaining paper is as follows: The proposed methodological framework is presented in detail in Section 2, while Section 3 is focused on its application and the presentation and analysis of the results. Conclusions and managerial implications are presented in Section 4.

## 2. Proposed Methodological Framework

### 2.1. Literature Review

Both policymakers and researchers have shifted their attention towards mobility plans and inquiries on how to best design and implement them. More recently, several further studies related to classification approaches, with respect to mobility actions and policies, have followed: Ranieri et al. [3] focused on the concept of "last mile logistics" in the urban areas by identifying collaborative and cooperative urban logistics, optimization of transport management and routing, and innovations in public policies and infrastructures as some of the main innovative contributions on which the current urban mobility field should focus on. Klecha et al. [4] analyzed information technology, such as mobile devices, and how they can be implemented as behavior-change strategies to promote citizen participation in the development process of the interventions. The authors discovered that there are still various unexplored possibilities for improving the situation when coping with challenges imposed by growing urbanization, thus, strongly encouraging further studies, especially in the fields of reflective learning and citizen involvement through "participatory methods". Ferrero et al. [5] studied car-sharing services and concluded that the development of ICT (information and communication technology) allows for a wide market penetration of new car-sharing models, including free floating car sharing services. Gallo & Marinelli [6] offered a review of the main actions and policies that can be implemented to promote sustainable mobility, aggregating them into three broad categories: (i) environmental, (ii) socio-economic, and (iii) technological, by concluding that "the problem is of interdisciplinary nature" and that "the achievement of sustainable mobility objectives requires different skills" which could be accumulated by integrating the knowledge and contribution from various stakeholders. The above recommendations have thus inspired the following study.

In the context of the effective and wide diffusion of sustainable mobility practices, Werland [7] provides several ways through which "sustainable planning" may be diffused within the EU, in particular by reviewing policy diffusion, policy transfer, and policy governance. Cao and McHugh [8] underline another significant aspect, which is the need for a systemic approach to the management of change. They underline that, in order to be effective, organizational change must be managed systematically, otherwise there is a great risk of failure [9–11]. In their work, it is emphasized that any change in management should be carried out gradually and with a clear vision and precision.

Hence, three general areas of interest emerge: Firstly, how to involve a wide array of stakeholders in the design and implementation of mobility plans in an even more effective way. Moreover, how to employ the most effective means of communication in order to reach those stakeholders and at the same time effectively coordinate them, and finally, how to view the entire process from a systemic perspective.

For that reason, the general objective of the proposed methodological framework is to classify the marketing techniques according to several relevant attributes and classes per step of the SUMP cycle. In particular, it aims to identify which marketing techniques best cope with regional needs, participation, and engagement, taking into account several contextual characteristics such as: the phase of the sustainable urban mobility plan development or implementation (existing phase in the SUMP cycle); the experience/maturity in sustainable planning and implementation; the maturity in terms of technical and administrative capability, and/or in terms of development and implementation of sustainable mobility plans.

This inductive analysis is therefore guided by a systemic approach. In fact, to have a deeper understanding of how to effectively handle the engagement and co-planning challenges of Sustainable Urban Mobility Planning and, more specifically, how to design acceptable policies for improving the adoption of more sustainable mobility modes, it is imperative to understand which dynamics together characterize the application of contexts and the techniques and which are the key success factors for those policies already in place. Rather than pinpointing the single causal chains leading to a favorable condition, the proposed approach aims to identify the dynamics enhancing the adoption of sustainable mobility in a specific context by addressing the plausible interrelationships between underlying factors.

Hence, the categorization process has been articulated in two phases: (1) systemic analysis and (2) systems factors concept mapping. In particular, the methodologies adopted to perform those analyses envisaged by these two phases, i.e., Systems Thinking evaluation approach and Group Concept Mapping (GCM), are briefly described in the next sections.

### 2.2. Systems Thinking (ST) and Causal Loop Diagrams

Systems Thinking [12] is an intellectual approach to reality that is intended to look at occurrences from a systemic perspective. This holistic viewpoint involves the analysis of not only distinct elements that an entity under study is composed of, but also of the intrinsic relationships between them.

In fact, a system is an interconnected set of elements that is coherently organized in a way that achieves a purpose. As it emerges from the previous definition, there are three components of a system [13]:

1. Elements: the entities which make up the system, they represent its fundamental constituents.
2. Interconnections: the relationships that link elements among each other. The structure of relationships defines a system as well as its elements; for example, the nature of the system football team does not vary even if all the members are changed. If instead interconnections are modified (for example rules are distorted), the nature of the football team changes.
3. Purpose: the goal which associates all the elements; without a purpose, a system loses its identity.

All systems are part of bigger systems, that in turn are part of even bigger systems, and so on, and in turn, are made up of sub-systems, that in turn are made up of their sub-sub-systems. A system is more than the sum of its components; this means that for understanding it, knowing the components is not sufficient, but a complete mapping of interconnections is needed. To do that, it is useful to notice that many of the interconnections in systems operate through flows of information [14,15].

More specifically, there are some factors (which occur frequently) that give rise to counterintuitive behaviors. There are mainly three factors:

- The misperception of delays;
- The misperception of feedback;
- Resistance to policies.

Thus, complex behavior arises because of the interaction of different feedback processes, and, of course, because of the presence of differential relations, non-linearity and delays. Systems Thinking is placed within this complex context and tries to give useful guidelines to address every aspect. In practice, as already mentioned, the approach of systems thinking is fundamentally different from that of traditional forms of analysis.

Following this point of view, in general, it can be argued that all the dynamics arise from two different feedback loops, namely the reinforcing feedback loops (positive feedback) and the balancing feedback loops (negative feedback). A feedback loop is defined as a close sequence of causes and effects, that is to say, a close path made of actions and information. Moreover, such a discipline characterizes the so-called Learning Organizations, namely the organizations that learn [16]. In particular, this discipline allows us to set aside all the incorrect mental models that cause a distorted and prejudicial vision of reality raising in this way a barrier to the learning process, which can instead develop when there is a common commitment.

The Systems Thinking approach employs various tools for extrapolating information about complex systems and discovering hidden and counter-intuitive behavior. One of the operational branches of Systems Thinking is System Dynamics. System Dynamics employ both qualitative and quantitative techniques to illustrate and simulate the behavior of systems over time. In this sense, the Causal Loop Diagram (CLD) instrument [17,18], typical of the System Dynamics approach, is heavily qualitative but is the starting point for the production of a quantitative model. Notwithstanding its qualitative value, the analysis of CLDs can introduce several important results [19]. The main advantage of using this type of analysis is that it provides a vision that considers many themes inside a system as interconnected with each other, contrary to those past approaches where systems were analyzed individually and on a sectoral basis. The advantages of Systems Thinking and System Dynamics are made compelling enough to be used in modeling the transportation sector (and all its various facets) including mobility plans [20–23].

*2.3. Group Concept Mapping*

A complementary perspective, allowing the understanding of the issue in its entirety as it provides a more bottom-up approach with its inductive reasoning, Group Concept Mapping is a mixed-methods strategy that captures the rich conceptual data from communities of interest on a particular question or topic and organizes and analyses it statistically using multidimensional scaling and cluster analysis. It involves a structured multi-step process [24] including, among others, data gathering, sorting and rating, clustering, etc., and results in the generation and interpretation of multiple maps. The process typically requires the identification of a large set of statements (or performance judgments) relevant to the topic of interest, individually sorting these statements into similar piles, rating each statement on one or more dimensions, and interpreting the maps that result from the data analyses. The analyses typically include multidimensional scaling (MDS) of the sorted data, clustering analysis of the MDS coordinates, and computation of average ratings for each statement and cluster of statements. The maps that result show the individual statements in two-dimensional (x, y) space with more similar statements located nearer each other and

grouped into clusters. Concept mapping has been used effectively to address substantive issues across a wide range of fields [25,26]. This process generates a conceptual framework for evaluation that has several benefits compared with less sophisticated conceptualization approaches such as focus groups:

- It represents a systematic process that integrates structured group processes such as unstructured idea sorting and rating tasks with sophisticated multivariate statistical methods to produce a well-defined, quantitative set of results;
- It graphically represents a domain of ideas in a framework that can be utilized directly for developing specific evaluation metrics;
- It facilitates the collection of input from a broad and diverse array of data sources in virtually any setting in which a group issue or need requires definition and evaluation, and it enables feedback on these data to participants in a timely manner [27].

Above all, concept mapping has been proven to be a valuable strategy to evaluate the results of practice-based research [28,29]. Using the concept map as a foundation, one can measure any number of variables of interest and display them as patterns on the map. Two or more patterns can be compared, both in the aggregate and in their details, using pattern matching to look at consensus and consistency over time, along with bivariate displays known as "go zones" to identify the potential courses of action or types of measurement. Therefore, in the context of the current paper, the method was modified adding to the extensive literature on the method's modifications as described by Rosas [30]. To the best of our knowledge, this is the first time that the method is used in such a way in the general transportation sector, and this can be considered a novelty of the current paper. The proposed methodological flow is:

**Step 1:** The analysis started by working on the data of the collected good practices. First, each of those practices was categorized by defining its objectives: the main one plus, if any, a secondary one. To improve the categorization effectiveness, and the opportunity to perform some comparisons, objectives were also standardized, and the good practices were separated into two general groups: those that aim at behavioral change towards sustainable mobility and those that aim at increasing co-planning levels.

**Step 2:** Then, within each group, each practice was matched with the SUMP cycle phases and steps (where the practice could be applied) defined by the European Guidelines for Developing and Implementing a Sustainable Urban Mobility Plan.

**Step 3:** Furthermore, each good practice was matched with the wider marketing technique category. The marketing techniques that were identified are described in Appendix A.

**Step 4:** The methods were adopted to implement the techniques, used to further classify the practices. The methods are described in Appendix A.

In order to provide a better understanding of those impacts produced by the implementation of techniques/methods, the combination of more than a single technique/method has also been considered and evaluated if observed in good practice.

**Step 5:** The obtained classification was then further improved by defining a set of relevant attributes and features of the practices, to investigate both the commonalities of them and those factors that could reveal determinants to understand their success. These features were:

- The Duration of the campaign: expressed as Continuous, Periodic, or One-time;
- The Locus of the events: expressed as Local, Regional, or National;
- The Cost implications: expressed as Low, Medium, and High;
- The Easiness to transfer: expressed as Low, Medium, and High.

**Step 6:** Finally, the success of a practice (thus, of combined techniques/methods implemented in a specific context) was measured against some Key Performance Indicators (KPIs). Such indicators are either explicitly mentioned in the description of the good practices or are inferred from their characteristics. A list of such KPIs follows for each group of good practices:

- GPs aimed at behavioral change:
  - a. People/stakeholders engaged/informed;
  - b. Citizens/stakeholders using (more) sustainable or energy-efficient modes of transport;
  - c. People registered in new mobility schemes.

- GPs aimed at co-planning:
  - d. People engaged/participating;
  - e. New mobility solutions/ideas co-created;
  - f. Level of people registered in a new mobility scheme.

**Step 7:** For each combination of practice/techniques/methods/SUMP steps, 6 specific *fitness indicators* were developed (The fitness indicators are provided in Appendix A).

Then, the K-Means [31] clustering technique was used. It is the most adopted in the GCM approach. In more detail, the algorithm was performed with 4 clusters, using the Euclidean distance and a random initial state. Finally, it should be stated that the technique was applied when 5 or more good practices were present in a specific SUMP step; any less would not provide meaningful calculations [32].

**Step 8:** To improve the understanding provided by the clustering step, first, an unstructured sorting [28] was applied to the practices by grouping them into piles, then both the practices and the techniques were rated according to a multi-criteria evaluation approach. In particular, 3 criteria were defined according to the indicators built in the previous step:

- Cost implications: The criterion requires minimization (the lower the value the better) and it can take the following values:
  - g. low, 1;
  - h. medium 2;
  - i. high 3.

- Easiness to transfer: The criterion requires maximization (the higher the value the better) and it can take the following values:
  - j. Low, 1;
  - k. Medium, 2;
  - l. High, 3.

- Impact on KPIs: The criterion requires maximization, and it can take the following values:
  - m. If all three attributes have values, 5;
  - n. If two of the attributes have values, 4;
  - o. With one attribute:
    - i. <1000 persons/stakeholders etc., value 1;
    - ii. >1000 persons/stakeholders etc., value 2;
    - iii. Any number of organizations, municipalities, etc., value 3.
  - p. No values, 0.

The rating was established by performing the TOPSIS ("Technique of Order Preference Similarity to the Ideal Solution") [33] Multi-Criteria Decision Analysis method. TOPSIS is considered one of the most versatile and easy-to-use MCDA method [34] and provided many valuable insights into the practices and marketing techniques (and related methods) [27]. Figure 1 summarizes the steps that are followed within the proposed framework.

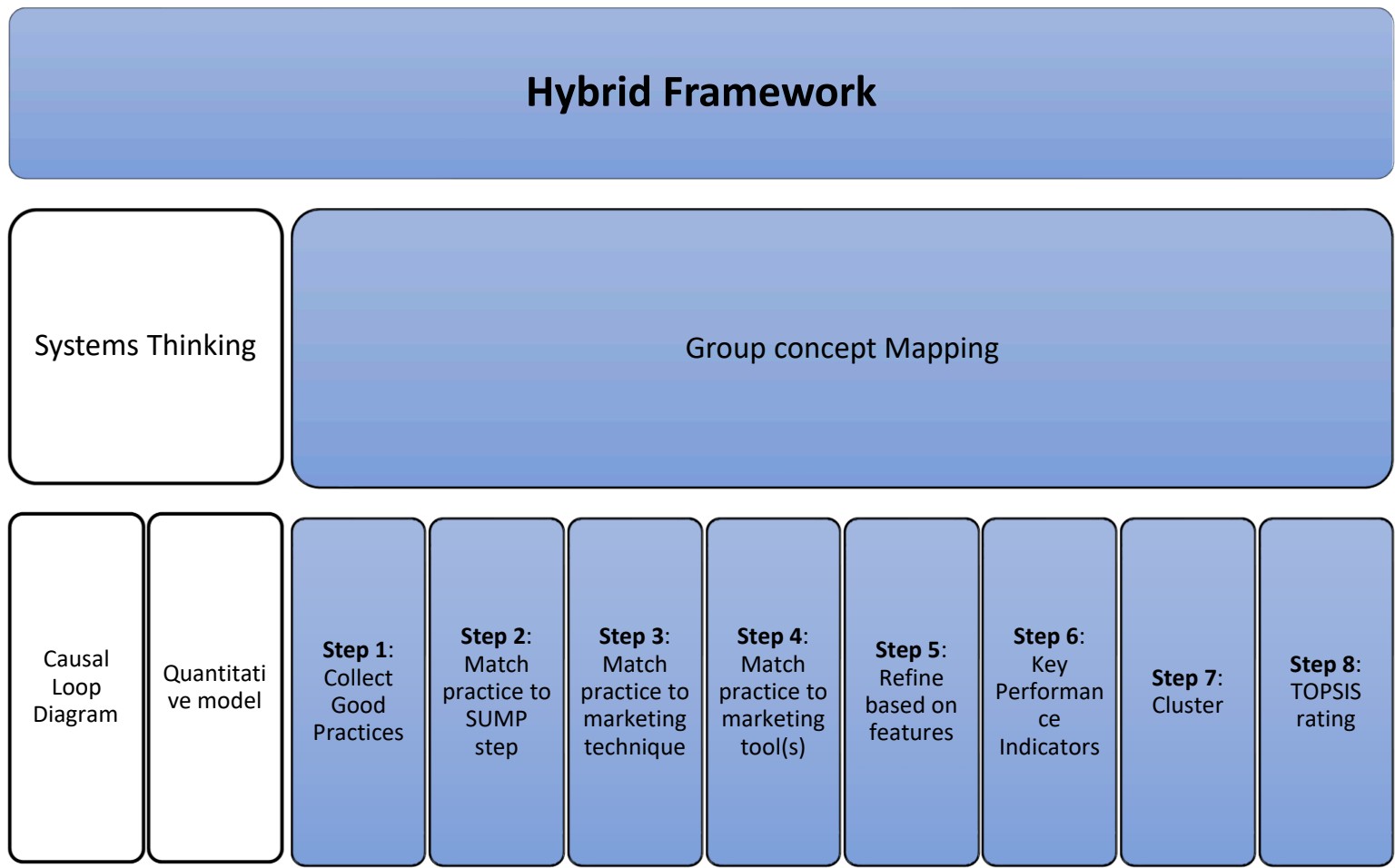

**Figure 1.** Graphical representation of the proposed framework.

## 3. Results

### 3.1. Results from the Systemic Analysis

In order to motivate citizens to accept one or more sustainable mobility solutions implemented in the city, it is necessary to understand that the adoption of a particular technology, product, or service passes through different stages. Each stage represents the current state of mind of the person who will use the technology, product, or service.

Citizens are, de facto, consumers of the city's services. Therefore, the process with which a citizen approaches the new sustainable mobility solution, an unusual means of transportation for him/her, can be analyzed as the process through which a consumer recognizes his/her need for that specific technology, product, or service; develops the willingness to adopt it by collecting information about it; and, finally, embrace totally the new technology, product, or service.

The Causal Loop Diagram (CLD) proposed in this section attempts to explain how this adoption process can be influenced by marketing tools and what the endogenous main variables that play a crucial role inside the process are.

In other words, this model aims to:

- Study the effectiveness of exploiting marketing tools for promoting sustainable mobility interventions among the population and for enhancing the participatory approach of SUMP;
- Investigate how marketing tools influence the adoption of new sustainable mobility interventions and whether specific characteristics of citizens should be taken into account when applying a marketing technique;
- Give evidence of how the adoption process, if well managed by policymakers, heavily affects the performance of the transportation system, as well as the health status of the city's environment (with obvious benefits for the city system in general)—serving sustainable mobility vision.

Figure 2 shows the model. Variables in green indicate the marketing techniques which enter the adoption process in different phases. Variables in red indicate the Key Performance Indicators which give relevant measures for the SUMP effectiveness.

The basic structure of the model was partially retrieved and adapted for our specific context from a model developed by Babader et al. [35]. In particular, the part transposed and adapted for our purposes is the backbone made by stocks and flows (uninformed -> informed -> aware -> practitioners).

The aim of this model is to clarify where and when the four relevant predictors were previously identified, namely: (a) the attitude towards the environment; (b) social pressure; (c) perception of policy effectiveness; (d) perceived benefits of the provided service, fit in the adoption process by citizens. To do so, as suggested by Babader et al. [30], the urban population was divided into different clusters (stocks), and then the predictors were factored in as so to be influencing the rates that determine the flows among the stocks.

As illustrated in Figure 2, the model concentrates on three main flows that define the rate with which the citizens change attitude (in favor of) towards SUMP process and towards sustainable mobility measures: (1) social and environmental concern, (2) SUMP solutions awareness, and (3) behavioral adaptation to SUMP solutions.

Concerning the first mentioned flow, the model identifies the variables that affect unconcerned citizens with regards to sustainable mobility, making them interested in the topic and willing to learn more about what their city has to offer. This change can be achieved by two main activities: on the one hand, by enhancing general social and environmental concerns; on the other, by pushing peer information sharing between citizens who are not concerned with the ones who do. Marketing techniques, in this sense, are valid tools to influence a target group to accept, reject, modify or abandon voluntary behavior, with the purpose of obtaining an advantage for individuals, groups or society as a whole. Therefore, the marketing techniques were included inside the model and were linked to specific dynamics in a logical manner, relying on the descriptions of the marketing techniques [27].

The activity related to enhancing general social and environmental concerns is related to psychological factors that arise from the repetitive behavior held by citizens. In other words, citizens generally have a certain resistance to adopting new solutions because they are stuck in habits that maximize their own individual short-term benefit, making the perception of long-term collective disadvantages even lower. The repetitive habits have a negative effect on the intention to change, slowing down the social and environmental process. The intention to change can be enhanced by increasing the social pressure regarding new sustainable thinking. To do so, working on social responsibility enhancement of the population through cause marketing, a marketing technique that focuses on social or charitable causes promoting social responsibility, seems to be a good solution. In addition, by highlighting environmental health status and problems to raise environmental concerns among the population, in this sense, the wheel of persuasion can also be a valid tool, as it is effective for persuading the targeted audience to embrace a new point of view through the psychology of conversion (insights from behavioral economics, consumer psychology, neuromarketing, sociology).

The other activity, i.e., to increase the flow from unconcerned to concerned citizens, is related to the information exchange between citizens who are not concerned with the ones who do. This can happen both in real life when citizens meet each other and virtually through social media and digital platforms (enhanced by "ad hoc" digital marketing, eventually). In order to boost information sharing and to give talking points to citizens, policymakers can leverage two other marketing tools: (i) undercover marketing, a form of marketing that uses 'sublime' messaging to promote a concept. It bears many similarities with the Word-of-Mouth technique as its objective is to create a "buzz" over a specific issue, (ii) guerilla marketing, an advertisement strategy that uses surprise and unconventional interactions in order to promote a concept.

After that, the model continues the investigation of what makes citizens become aware of an SUMP policy in their city. The model identifies, similarly to the previous case, that the information-sharing process between citizens is one of the ways through which uninformed citizens become aware of SUMP scopes currently underway inside the city. In this specific case, the most suitable marketing tool to enhance the SUMP policy discovering process is dialogue marketing, which uses technological advancements such as personalized websites, social media apps, and blog platforms to promote a message focusing on those individuals who are already interested and open to engagement and creates opportunities for them to connect and relate. There is also digital marketing that can help this flow, as for the previous one, because it utilizes the internet and online-based digital technologies such as desktop and mobile media, digital apps, and other platforms to promote services and products. In this way, citizens who want to search for information independently from others can consult the material uploaded by SUMP marketing operators directly on the web.

The last stage in the model is to investigate people's behavioral adaptation to become practitioners of SUMP solutions. This is the hardest state to be reached because, although a citizen can be concerned about the environment and be informed about solutions, he/she could not want to enjoy or adopt the solution because of insecurities about the benefits he/she can get from it, in line with the "resistance to adopt" factor, of which we have spoken before. The resistance can be tackled by positive word-of-mouth about the SUMP policy. In fact, the most effective way citizens could be persuaded to use a SUMP solution is when they are pushed by someone that they know and whose opinion they trust. The policymaker can boost the word-of-mouth phenomenon by investing in public events, as well as public relations activities. There is, however, a critical aspect to consider in this dynamic. The opinion about the policy of the person who is spreading the message must be positive, otherwise the effect would be diametrically opposed. In order to improve the opinion about the solution, and for it to be effective, the more the practitioners of the solution the more the solution is effective and successful. The effectiveness has two beneficial effects. On the one hand, it could convince someone to try that solution because he/she has seen that there are many citizens already doing so. On the other hand, in the

long term the effectiveness involves a benefit for public savings, that involves savings of citizen taxes and at the same time causes satisfaction in practitioners about policy makers' work as they see its virtuosity, stimulating positive word of mouth about SUMP. It is worth mentioning that the satisfaction of practitioners can also be enhanced by using relationship marketing activities, as it emphasizes specific target groups with the intention of building long-lasting relations through a form of communication that is extended beyond just informing (Figure 3) [27].

In the following paragraphs, some of the most important feedback loops of the model will be examined in detail as well as what insights they provide for managers and policymakers.

As depicted in Figure 4, during each phase in which citizens develop new features towards the reach of "practitioner" status, there are several feedback loops that are presented in pairs (B1 and R1, B2 and R2, B3 and R3). They represent the dynamic of saturation in different phases. In fact, taking the first combination as an example, while there is a reinforcing loop R1 that eventually increases the number of citizens concerned with sustainable mobility, on the other hand, the balancing loop B1 prevents this growth from being excessive because as the concerned citizens raise in number, the opportunity for them to encounter unconcerned citizens becomes lower, until reaching the saturation, thus the equilibrium. This is true for all three dynamics shown.

The loops of Figure 4 give important indications about how, and especially when, it is better to use a particular type of marketing technique. Undercover marketing and guerrilla marketing fit better with SUMP strategy when there is still the need to develop a common ground of social and environmental responsibility among citizens. These two kinds of tools act like kick-starters for information sharing about transport and environmental issues, pushing citizens to take care of such topics. Once almost all citizens have received this "education" about the topics, it seems that it would be more convenient to invest in other marketing types, for example, dialogue marketing if there are many citizens who do not know SUMP's scopes and the role is undertaken by their city; or word of mouth marketing when the SUMP solutions need a catalyst to spread among the population [27].

As depicted in Figure 5, besides the phenomena of information sharing and social influence that are typical dynamics inside a population, there are also other loops equally important.

The reinforcing loop called R4 (loop in orange) is related to the dynamic explained before about being blocked in habits that maximize one's own individual short-term advantage, the more the individual advantages the more repetitive behaviors are held by citizens. This loop stays inside another main loop, i.e., R5 (loop in blue), affecting its behavior. The R5 loop is quite dangerous because, having R4 inside, it is likely to grow too much out of control due to the selfish tendency of citizens; when this happens, the social responsibility of people falls along with their intention to change, fueling the resistance to adopt new habits. Another loop to be considered is the R6 loop (green). If the citizens are not convinced about the utility of SUMP and sustainable mobility measures they can receive in the long term, they prefer to continue to receive short-term advantages through repetitive irresponsible behaviors. Conversely, if the resistance to adopt is decreasing, R6 becomes a virtuous circle through which citizens recognize the value of the policy (since everyone is using it), and the policy will be adopted more and more easily over time, becoming common practice.

Lastly, there is a fundamental balancing loop, B4 (in red), which prevents the aforementioned reinforcing loops from becoming uncontrollable once they start to increase or decrease. In this loop, the environmental health status represents the balance needle of the system, through which the citizens feel that there is something wrong with their behavior. Therefore, when the environmental health drops down very quickly and the consequences of such a state come out, there is a reaction of environmental concern from citizens that increase the attitude of citizens and their everyday sustainable practices, with benefits for overall social responsibility that pushes, in turn, the social pressure towards change. This whole dynamic helps to tackle the resistance to adopt, with the consequent benefits.

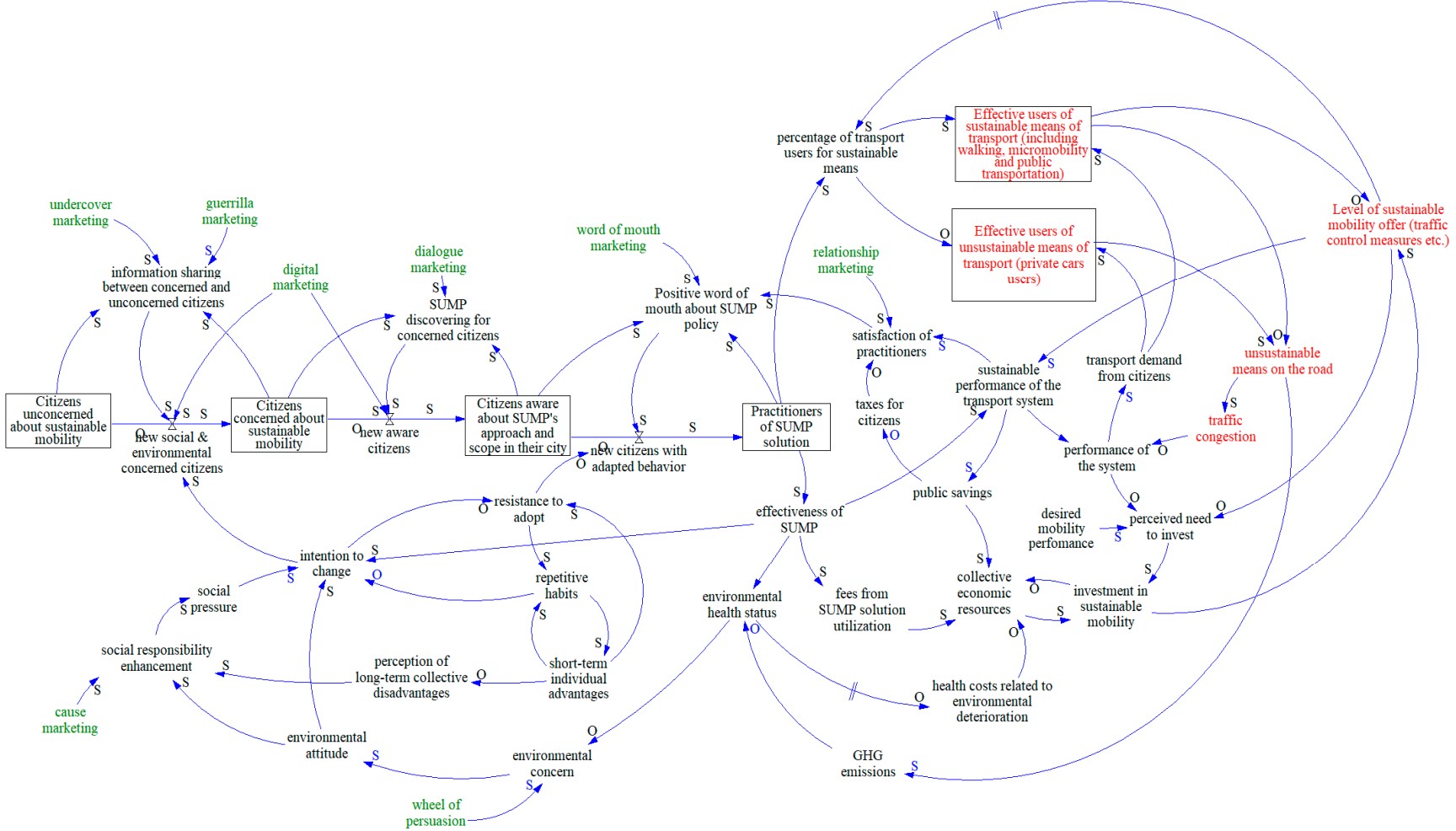

**Figure 2.** Causal Loop Diagram.

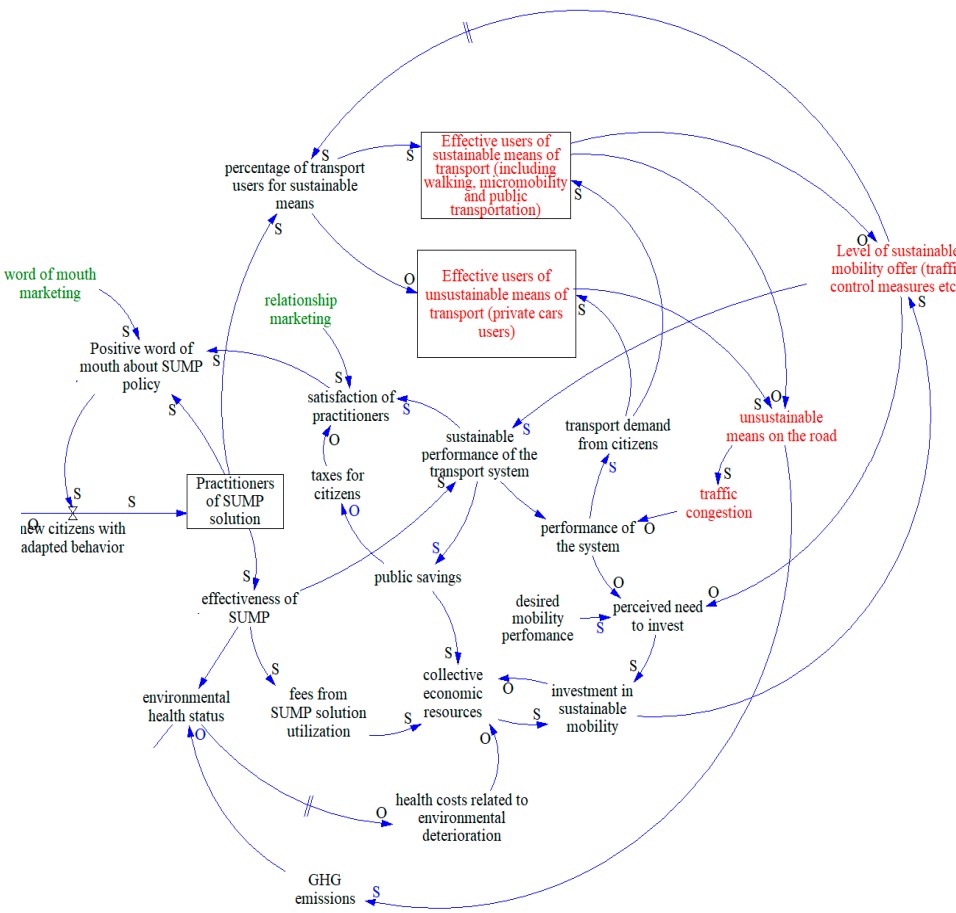

**Figure 3.** Increasing practitioners and the system's performance dynamics.

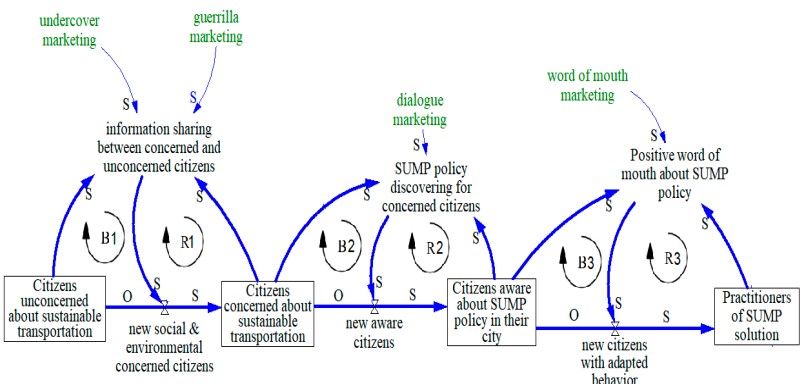

**Figure 4.** Feedback loops through adoption process.

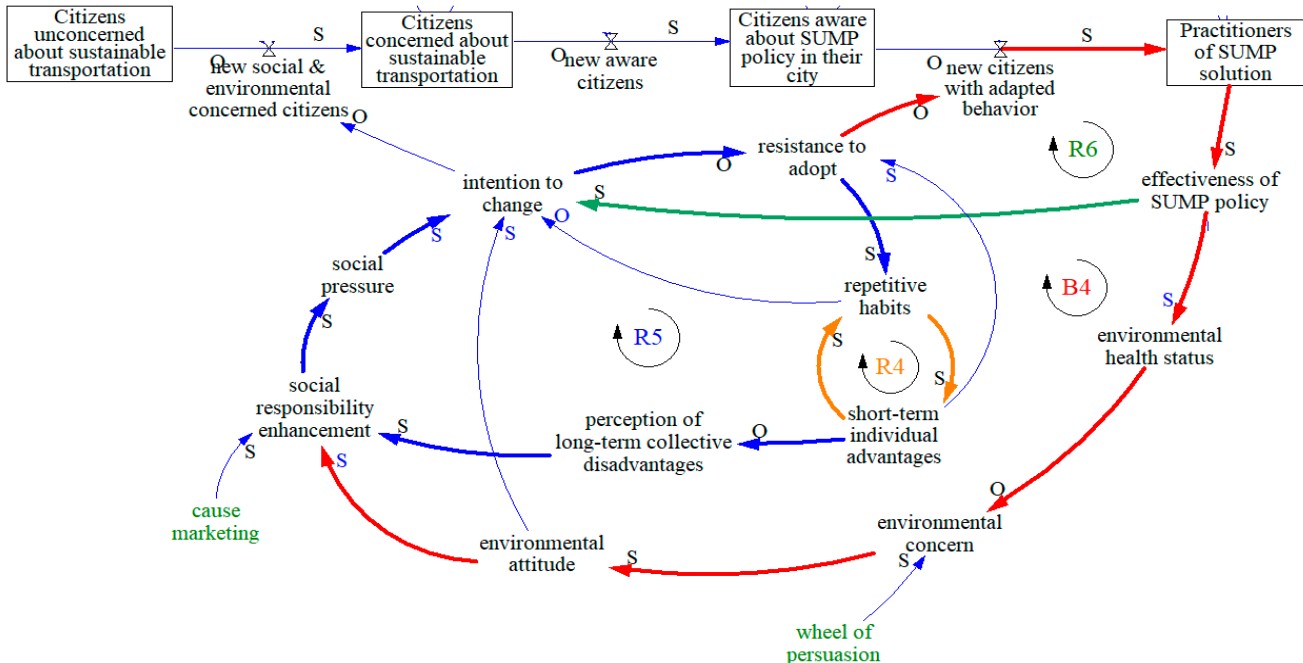

**Figure 5.** Feedback loops through the adoption process.

Hence, resistance to adopt is a crucial variable inside the presented process, through which all the loops, directly or indirectly, pass, but there is something that policymakers could do to control it. First of all, when the sense of community is lacking, policymakers could act on citizens' social responsibility through cause marketing. By doing this, the R5 loop (which is one of the most dangerous) can be brought under control with evident benefits on resistance to adoption (avoiding at the same time the negative consequences of the R4 loop). Another means to tackle the citizens' resistance to adopt is to work on the flow named "new citizens with adapted behavior" through enhancing word-of-mouth marketing. In this way, the practitioners will increase naturally, and simultaneously they will give evidence that the SUMP solutions are effectively working and this will push undecided citizens to take a try. The last action that can be taken is about loop B4. Generally, this loop is slow to occur because there is a certain delay between the environmental health status and the negative consequences on people, so the awareness and concern about the problem come after some time. In order to speed up the process, policymakers could make use of the wheel of persuasion to effectively persuade the targeted audience to focus on environmental issues before the potential consequences come out.

Other policy recommendations that are revealed in the CLD include the fact that the larger the number of users of sustainable means of transportation will (in general) result in a reinforcing loop that will increase the number of users itself. As a result, any policy maker should bear in mind that increasing the number of commuters that adopt a sustainable means of transportation can have positive effects (even if they are small) on the overall sustainability of the urban environment, as the increased number of such users acts in a reinforcing way to itself. However, to increase the probability of success, the increase in those users should be accompanied by an increased provision of sustainable mobility options (quality, coverage, accessibility), otherwise, the balancing loop B8 mentioned in a previous paragraph could have the opposite effect and significantly reduce the number of users of sustainable means.

Furthermore, in general, the central economic/financial variables act in such a way that restrains the number of users of sustainable means and/or reinforces the number of users of unsustainable means. Consequently, the shift to a sustainable means of transportation requires constant resources (i.e., maintenance and enforcement costs other than initial investment costs) by the appropriate authority. Moreover, a successful marketing technique

(relationship marketing and word-of-mouth) will act in the feedback loops by increasing the number of adopters. This will result in increased effectiveness of the policy. Thus, the importance of marketing is that, if successful, it can render the balancing economic loops into reinforcing ones, increasing the probability of a sustainable transition without draining the collective economic resources.

Finally, the CLD illustrates that the environmental variables act in general in a positive/reinforcing way to the other variables and loops of the system. This means that when the environmental status is at satisfying/good levels, then the overall performance and sustainability of the system become even better. However, when the environmental status is not at good/satisfactory levels, then the performance of the system falls rapidly, which results in an even bigger environmental degradation. These effects nonetheless are not immediately observed in the system because these loops contain some significant delays, with the most important being the one connecting the environmental health status with the health costs related to environmental deterioration. This delay is extremely uncertain and can span from months to years. As a result, the negative effects can manifest later in time. This danger could explain why environmental deterioration may be underrepresented and/or underappreciated in policy-making [27].

*3.2. Results from the Modified Group Concept Mapping Approach*

The second part of the proposed methodological framework employs Group Concept Mapping on the good practices according to the steps that were mentioned in Section 2.3. Due to the large part of good practices and data that were collected, only the set of practices that aims at behavioral change and are placed on SUMP step 11 will be analyzed. Furthermore, due to data ownership, the good practice will be recognized only by an ID number. However, all the data can be made available after request. The details of the good practices are displayed in Appendix B.

There were 25 Good Practices (GPs) that are matched with the SUMP step 11, 4 with the SUMP step 10, 2 with the SUMP step 3, 1 with the SUMP step 4, and 1 with the SUMP step 12. As a result, while we could argue that the techniques applied to enhance a behavioral change in the contextual SUMP steps 10, 3, 4, and 12, are the most effective ones, further cluster analysis can be performed for those GPs that match with the SUMP step 11. Such an investigative analysis has been performed using the following practices' main factors (i.e., main features and attributes, Table 1).

The first cluster contains the practices with the following IDs: 7, 11, 12, 21, 26, 27, 32, 35, and 36. The majority of those practices use a combination of marketing techniques, while almost half of them are "continuous". Finally and most importantly, it appears that practices that belong to this cluster are either "local" (the majority of them) or at most "regional" in location. Consequently, it can be induced that: when policymakers want to enhance public awareness for sustainable mobility interventions (SUMP step 11), they are usually applying a combination of marketing techniques with a combination of diverse communication approaches. This can be explained due to the crucial role of well-communicating mobility interventions scopes (and services details, opportunities, coverage, and the role within the wider city's vision) when they started being provided to citizens.

The second cluster contains the practices with the following IDs: 1, 10, 13, 20, 22, 30, 38, and 43. These practices do not have many commonalities, thus, further analysis is required to extract robust/safe conclusions.

The third cluster contains the practices with the following IDs: 2, 5, 18, 25, 31, and 39. The practices have a "continuous duration" and are either "regional" or national. Within this cluster, "digital marketing" and "wheel of persuasion" are the dominant marketing techniques. Consequently, it is a common practice to activate digital marketing or wheel of persuasion as marketing techniques during phase 4 and step 11 of the SUMP cycle. E-engagement seems to be a strong marketing tool—respecting also COVID-19 social distancing measures.

Finally, the fourth cluster contains only the practices with the IDs 23 and 37, thus its size prevents a safe extraction of conclusions about the marketing techniques without a further analysis.

To gain further insights, a ranking of all the practices was devised adopting the TOPSIS method. In particular, three features and attributes related to the practices were adopted to perform such an analysis: (1) cost implications, (2) easiness to transfer, (3) impact on KPIs. Finally, the ranking was elaborated only for those sets of practices matching with a specific SUMP step with a meaningful size: the practices matching with SUMP cycle step 11 and the ones matching with the SUMP cycle step 10.

As concerns the first set (i.e., practices matching with the SUMP cycle step 11), the TOPSIS rating is represented in the following Figure 6 [27].

The best eight rated practices (11, 12, 43, 13, 1, 20, 25, and 27) have a low-to-medium cost and a medium-to-high transferability barrier, and their attributes, as reported in Table 2, suggest that policymakers aiming at a behavioral change in SUMP step 11 should adopt a combination of several marketing techniques, mostly the "digital marketing" and the "word of mouth" ones.

In conclusion, the combination of both methodological frameworks provided several insights into when it is appropriate to use each marketing technique in sustainable mobility plans. Table 2 summarizes those insights below.

**Table 1.** Attributes and values to classify the practices matching with the SUMP step 11.

| ID | Duration of the Campaign | Marketing Communication Techniques | Diversity of Communication Methods | Locus of Events | Cost Implications | Easiness to Transfer | Impact on KPIs |
|---|---|---|---|---|---|---|---|
| 1 | Periodic (2) | Word of Mouth (6) | Public event (6) | Local (1) | Low (1) | High (3) | 2 |
| 2 | Continuous (3) | Digital Marketing (4) | E-engagement campaign (9) | Regional (2) | High (3) | Low (1) | 2 |
| 5 | Continuous (3) | Word of Mouth (6) | Ambassador campaign (12) | Regional (2) | Medium (2) | High (3) | 2 |
| 7 | Periodic (2) | Cause Marketing (8) | Public cause event (1) | Local (1) | Low (1) | High (3) | 1 |
| 10 | No Information (0) | Word of Mouth (6) | Other (3) | Local (1) | High (3) | Low (1) | 2 |
| 11 | No Information (0) | Combined (10) | Awareness campaign (4) | Local (1) | Medium (2) | Medium (2) | 5 |
| 12 | No Information (0) | Combined (10) | Combined (2) | Local (1) | Medium (2) | Medium (2) | 4 |
| 13 | No Information (0) | Wheel of Persuasion (2) | Combined (2) | Local (1) | High (3) | Medium (2) | 4 |
| 18 | Continuous (3) | Wheel of Persuasion (2) | Capacity building (7) | National (3) | Medium (2) | Low (1) | 2 |
| 20 | Continuous (3) | Word of Mouth (6) | Combined (2) | National (3) | Low (1) | High (3) | 2 |
| 21 | Continuous (3) | Combined (10) | Combined (2) | Local (1) | Medium (2) | High (3) | 2 |
| 22 | Periodic (2) | Guerilla Marketing (5) | Pilot intervention (5) | Local (1) | Medium (2) | High (3) | 0 |
| 23 | One time (1) | Undercover Marketing (7) | Popular event (8) | Local (1) | Medium (2) | Medium (2) | 0 |
| 25 | Continuous (3) | Digital Marketing (4) | E-engagement campaign (9) | Regional (2) | Low (1) | High (3) | 2 |
| 26 | Continuous (3) | Combined (10) | Combined (2) | Local (1) | High (3) | Medium (2) | 1 |
| 27 | Continuous (3) | Combined (10) | Combined (2) | Local (1) | Low (1) | High (3) | 2 |
| 30 | Continuous (3) | Undercover Marketing (7) | Other (3) | Regional (2) | Medium (2) | Medium (2) | 0 |

<div align="center">**Table 1.** *Cont.*</div>

| ID | Duration of the Campaign | Marketing Communication Techniques | Diversity of Communication Methods | Locus of Events | Cost Implications | Easiness to Transfer | Impact on KPIs |
|---|---|---|---|---|---|---|---|
| 31 | Continuous (3) | Digital Marketing (4) | E-engagement campaign (9) | Regional (2) | Low (1) | High (3) | 0 |
| 32 | Continuous (3) | Combined (10) | Combined (2) | Regional (2) | Medium (2) | Medium (2) | 0 |
| 35 | One time (1) | Combined (10) | Combined (2) | Local (1) | Low (1) | High (3) | 1 |
| 36 | Continuous (3) | Combined (10) | Combined (2) | Regional (2) | High (3) | Low (1) | 0 |
| 37 | Continuous (3) | Combined (10) | E-participation/crowdsourcing (10) | Regional (2) | Medium (2) | Medium (2) | 0 |
| 38 | Continuous (3) | Wheel of Persuasion (2) | Combined (2) | Regional (2) | High (3) | Medium (2) | 1 |
| 39 | Continuous (3) | Wheel of Persuasion (2) | Capacity building (7) | National (3) | Medium (2) | Medium (2) | 0 |
| 43 | Continuous (3) | Relationship Marketing (3) | Awareness campaign (4) | National (3) | Medium (2) | Medium (2) | 4 |

The clustering of practices according to these attributes, by means of the K-Means technique, resulted in 4 different clusters.

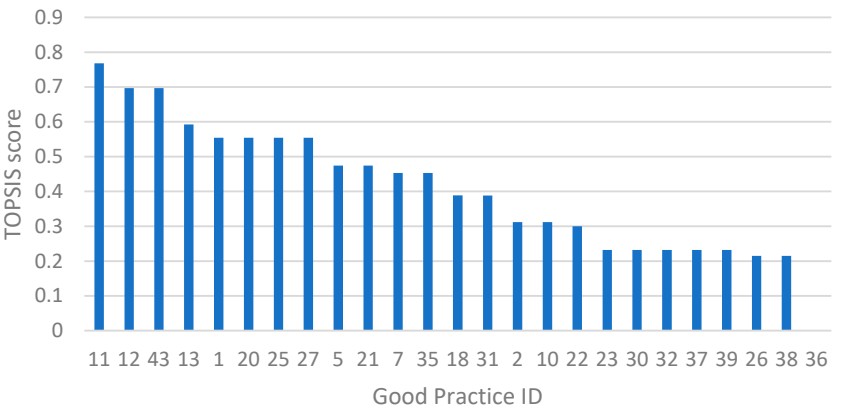

**Figure 6.** TOPSIS ranking of the practices matching with the SUMP cycle step 11.

**Table 2.** Marketing techniques and when it is appropriate to use them.

| Marketing Technique | When It Is Appropriate to Use It According to Systems Thinking and System Dynamics | When It Is Appropriate to Use It According to the Modified Group Concept Mapping Framework |
|---|---|---|
| Undercover marketing | When there is still the need to develop a common ground of social and environmental responsibility among citizens | |
| Guerrilla marketing | When there is still the need to develop a common ground of social and environmental responsibility among citizens | |
| Word-of-mouth | When the SUMP solutions need a catalyst to spread among the population. | |
| | Tackle the citizens' resistance to adopt | |
| | When there is the need to increase the number of adopters of sustainable means of transportation. | |
| | | Aim at behavioral change in SUMP step 11 |
| | | Policymakers addressing the SUMP cycle step 10 |

**Table 2.** *Cont.*

| Marketing Technique | When It Is Appropriate to Use It According to Systems Thinking and System Dynamics | When It Is Appropriate to Use It According to the Modified Group Concept Mapping Framework |
|---|---|---|
| Cause marketing | When the sense of community is lacking | |
| Wheel of persuasion | To effectively persuade the targeted audience to focus on environmental issues before the potential consequences come out | During phase 4 and step 11 of SUMP cycle |
| Relationship marketing | When there is the need to increase the number of adopters of sustainable means of transportation | |
| Digital marketing | | During phase 4 and step 11 of SUMP cycle |
| | | Aim at behavioral change in SUMP step 11 |
| Guerilla marketing | | Policymakers addressing the SUMP cycle step 4 |
| Combination of marketing techniques | | When policy makers want to enhance public awareness for sustainable mobility interventions |

## 4. Conclusions

In the previous sections, a categorization of the marketing techniques and methods have been produced, providing several insights for application and for understanding the dynamics underpinning the transition to a more sustainable mobility condition starting from a contextual-dependent SUMP cycle step. The categorization was based on a novel methodological framework that combines Systems Thinking (and System Dynamics) with a modified Group Concept Mapping approach.

Regarding the System Dynamics part, the analysis of the main variables and feedback loops of the (part) of the Causal Loop Diagram, revealed several insights that could be helpful to policymakers. In the big view drawn by the CLD model, the marketing techniques were considered as input levers for the policymakers to lead the system towards specific results in terms of performance of sustainable mobility. In this sense, marketing techniques are catalysts that should be used in line with their specific aims; in other words, each technique has its own role and timing in the adoption process (as clearly described inside the CLD) and this must be clear to policymakers who want to make use of them. Nevertheless, the right positioning of marketing techniques cannot be the only advice from this analysis, as there are also other aspects to be considered in the process which are relevant for the success of the entire decision-making process.

Increasing the number of users that are willing to adopt a sustainable means of transportation, can have positive effects (even if they are small) on the overall sustainability of the urban environment, as the increased number of such users acts in a reinforcing way to itself. However, to increase the probability of success, the increase in those users should be accompanied by an increase in the capacity of the transportation system to accommodate them, otherwise, the balancing loops have the opposite effect and significantly reduce the number of users of sustainable means. Causal, undercover, and guerilla marketing can support the startup phase of awareness raising while then, word-of-mouth can take the reins.

The environmental variables act in a positive/reinforcing way to the other variables and loops of the system. This means that when the environmental status is at satisfying/good levels (there is little or no environmental degradation), then the overall performance and sustainability of the system become even better. However, when the environmental status is not at good/satisfactory levels, then the performance of the system falls rapidly which results in an even bigger environmental degradation. These effects, nonetheless, are not immediately observed in the system, because these loops contain some significant delays, with the most important being the one connecting the environmental health status with the health costs related to environmental deterioration.

As for the modified Group Concept Mapping approach, with the clustering and ranking analysis of the good practices, the following insights were revealed:

Investigating those practices aimed at behavioral change, policymakers addressing the SUMP cycle step 11 should apply a combination of marketing techniques with a combination of diverse communication approaches while limiting the spatial scope to specific locations whether local or regional. Furthermore, they should use digital marketing or the wheel of persuasion with a focus on e-engagement at a regional or national level and with a big timeframe.

Furthermore, policymakers addressing the SUMP cycle step 10 should consider word-of-mouth as the best suitable marketing technique since it appears to be cost-effective and easy to transfer to different contexts while generating adequate/satisfying results.

On the other side, by investigating those practices aimed at co-planning, we learned that policymakers addressing the SUMP cycle step 3 could use a combination of marketing techniques; however, they should be continuous in time but local.

Finally, policymakers addressing the SUMP cycle step 4 could use Guerilla marketing with gaming but in a local context.

Future directions of the research include the expansion of the good practices to investigate whether the results hold when more data is available and apply the proposed framework to get updated insights. In addition, in the present work, we did not consider the different types of personalities or how they might react to different marketing techniques. Furthermore, the list of marketing techniques themselves was dictated only from the good practices that were studied. Thus, future directions of our research include studying these important aspects and how they might affect the results of the proposed framework. Moreover, a Graphical User Interface could greatly enhance the communication capabilities of the framework and allow end users to apply. Finally, the authors believe that the proposed framework could be used in settings other than transportation and mobility and plan to test their assumption.

**Author Contributions:** Conceptualization, S.A. and R.O.; methodology, R.O., S.A., A.P. and G.T. validation, R.M. and S.A.; data curation, G.T. and R.O.; writing—original draft preparation, R.M., G.T. and A.P.; writing—review and editing, S.A. and R.O. All authors have read and agreed to the published version of the manuscript.

**Funding:** This research was funded by the e-smartec project, a three-year project that started in August 2019, funded under the Interreg Europe Programme 2014–2020 and addressing the thematic area of Low Carbon Economy.

**Institutional Review Board Statement:** Not applicable.

**Informed Consent Statement:** Not applicable.

**Data Availability Statement:** Data are available upon request.

**Conflicts of Interest:** The authors declare no conflict of interest.

## Appendix A

Marketing techniques of step 3 of the proposed framework:

- Dialogue Marketing. The generic term for all marketing activities in which media is used with the intention of establishing an interactive relationship with individuals. Dialogue marketing uses technological advancements such as personalized websites, social media apps and blog platforms to promote a message focusing on those individuals who are already open to engagement and creates opportunities for them to connect and relate.
- Relationship marketing. Form of marketing that emphasizes in specific target groups with the intention of building long lasting relations. Communication is extended beyond informing. It can be interpreted as commodity exchange that instrumentalize features of partnership.
- Digital Marketing. The component of marketing that utilizes internet and online based digital technologies such as desktop and mobile media, digital apps and other

platforms to promote services and products. The same policies can be applied to participatory projects for citizens' engagement. Types of Digital marketing, basing on the primary means of communication that each campaign uses:

- Social Media. A technique that primarily uses blogs or communicative platforms such as Twitter, Facebook, Instagram, YouTube, Snapchat.
- Internet Marketing. A technique that primarily uses mobile and desktop media with platforms or Web based apps
- Word of mouth. A process of storytelling and knowledge spread from one person to another based on reputation. It is the most straight-forward technique since it is based on personal communication and recommendation. Its objective is to generate "buzz" over specific issues which in return will generate awareness and further participation in future initiatives.
- Undercover marketing. Form of marketing that uses sublime messaging to promote a concept. The audience is exposed favorably to a topic or issue without being aware of the promotion strategy. It bears many similarities with the Word of Mouth technique as its objective is to create a "buzz" over specific issue. What differentiate it is the use of alternative engaging methods.
- Guerrilla Marketing. An advertisement strategy which uses surprise and unconventional interactions in order to promote a concept.
- Wheel of persuasion. The technique where scientific insights on the psychology of conversion (insights from behavioral economics, consumer psychology, neuromarketing, sociology) are used for persuading the targeted audience.
- Cause marketing. The marketing technique that focuses on social or charitable causes promoting social responsibility. It is designed to raise attention around the topic, while linking relevant activities or ideas that can benefit the explored topic. Methods for marketing techniques of step 4 of the proposed framework:
- Surveys. Reaching audience through dedicated surveys and via personal interviews
- Focus groups. A combo method of focused interviews and a discussion group. It is designed to obtain information about (various) people's preferences and values on a defined topic.
- Public Consultation.  It is a public enquiry targeted to a group of randomly selected citizens.
- Experts Panels. A specialized discussion where a variety of experts is engaged; based on various fields of expertise; to debate and discuss various courses of action and make recommendations.
- Public Events. Events intended to raise awareness, by creating opportunities to inform the public about issues and projects that are being explored.
- Raising Awareness Campaign. A promotional campaign which uses several tools in order to reach as many individuals as possible.
- Workshop. An intensive planning session where citizens, designers and others collaborate on a vision for development.
- Participatory Mapping. A general term used to define a set of approaches and techniques that combines the tools of modern cartography with participatory methods to represent the spatial knowledge of local communities.
- e-Participation—crowdsourcing. E-participation is the utilization of information and communication technology in order to motivate and engage wider citizens through diverse modes of technical and communicative skills. An online tool which enables involvement in decision co-creation process, in various extent.
- e-Engagement—campaigning. Uses information technology (IT) and digital tools to facilitate the process of engagement.
- Gaming. A game is a simulation of a real situation, allowing participants to act out and experience interactions of community activities. It is a participatory approach to problem solving that engages a real-life situation compressed in time so that the essential characteristics of the problem are open to examination.

- Gamification. The use of game-elements in non-game contexts. It refers to an instructional strategy with the aim to increase engagement, motivation, and participation by integrating game strategies such as point scoring, competition features, rules of play, etc., to an online platform or community, or mobile application.
- Pilot Interventions. Is an approach were interventions of a temporary character are implemented on trial base, leading towards a more permanent transformation in the future.
- Capacity building. A method that develops further a certain range of skills and competencies of the participants.
- Popular Events. A method where well established events and happenings are "side" used.
- Ambassador campaign. Indirect promotion by collaborating important public figures (celebrities, opinion-leaders, trendsetters).
- Public Cause Events. Events that are dedicated to social or charitable causes.
- Raising Awareness campaign (Cause related). Awareness campaign that focuses on social and charitable causes. Fitness indicators of step 7 of the proposed framework:
- Duration of the campaign. A value to the duration of the campaign was attributed, with the purpose of facilitating the numerical calculations of the clustering process. The indicator takes one of the following values: 1 for One time; 2 for Periodic; 3 for Continuous.
- Marketing communication technique/s. indicator related to marketing technique mostly implemented in the practice. It takes the following values: Dialogue marketing, 1; Wheel of persuasion, 2; Relationship marketing, 3; Digital marketing, 4; Guerilla marketing, 5; Word of mouth, 6; Undercover marketing, 7; Cause marketing, 8; Other, 9; Combined (if the identification of a single main technique), 10.
- Means of communication. An indicator to attribute a value to the methods associated with a combination of practice/techniques:

  q. No information, 0
  r. Public cause event, 1
  s. Combined, 2
  t. Other, 3
  u. Awareness campaign, 4
  v. Pilot intervention, 5
  w. Public event, 6
  x. Capacity building, 7
  y. Popular event, 8
  z. E-engagement-campaigning, 9
  aa. E-participation/crowd [21–23] sourcing, 10
  bb. Gaming, 11
  cc. Ambassador campaign, 12

- Cost implications:

  dd. low, 1
  ee. medium 2
  ff. high 3

- Easiness to transfer:

  gg. Low, 1
  hh. Medium, 2
  ii. High, 3

- Impact on KPIs:

  jj. If all three attributes have values, 5
  kk. If two of the attributes have values, 4
  ll. With one attribute:

      i.      <1000 persons/stakeholders etc., value 1

      ii.     >1000 persons/stakeholders etc., value 2

      iii.    Any number of organizations, municipalities etc., value 3

mm.   No values, 0 [27]

## Appendix B

Table A1 below reports the practices aiming at behavioral change, along with the matching with the related SUMP cycle and steps. The nature of each practice may be such that it belongs to more than one cycles and more than one steps within that cycle, which underlines the complexity of the task that policy makers face and reinforces the usefulness of the proposed methodological framework.

**Table A1.** Practices aiming at behavioral change, their SUMP cycles, steps and attributes.

| Good Practice ID | *MAIN OBJECTIVE* | *SECONDARY OBJECTIVE* | SUMP Phase | SUMP Step | SUMP Phase | SUMP Step | SUMP Phase | SUMP Step |
|---|---|---|---|---|---|---|---|---|
| 1 | increase awareness regarding the benefits deriving from shifting to sustainable or green modes of transport | | 4 | 11 | 3 | 8;9 | | |
| 2 | exchanging information with citizens and stakeholders | increase awareness regarding the benefits deriving from shifting to sustainable or green modes of transport | 4 | 11 | | | | |
| 3 | exchanging information with citizens and stakeholders | increase co-creation of new mobility solutions/ideas (achieving co-creation) | 1 | 3 | 2 | 4;5;6 | 4 | 11;12 |
| 5 | increase awareness regarding the benefits deriving from shifting to sustainable or green modes of transport | | 4 | 11 | 3 | 8;9 | | |
| 7 | increase awareness regarding the benefits deriving from shifting to sustainable or green modes of transport | | 4 | 11 | 3 | 8;9 | | |
| 9 | increase co-creation of new mobility solutions/ideas (achieving co-creation) | increase awareness regarding the benefits deriving from shifting to sustainable or green modes of transport | 1 | 3 | 4 | 11 | 3 | 9 |
| 10 | increase awareness regarding the benefits deriving from shifting to sustainable or green modes of transport | | 4 | 11 | 3 | 8;9 | 1 | 3 |
| 11 | increase awareness regarding the benefits deriving from shifting to sustainable or green modes of transport | | 4 | 11 | | | | |
| 12 | increase awareness regarding the benefits deriving from shifting to sustainable or green modes of transport | | 4 | 11 | | | | |
| 13 | increase awareness regarding the benefits deriving from shifting to sustainable or green modes of transport | behavioural change towards sustainable or green modes of transport | 4 | 11 | | | | |
| 14 | increase awareness regarding the benefits deriving from shifting to sustainable or green modes of transport | behavioural change towards sustainable or green modes of transport | 4 | 10 | 3 | 8;9 | | |
| 15 | increase awareness regarding the benefits deriving from shifting to sustainable or green modes of transport | | 4 | 10 | 3 | 8;9 | | |
| 18 | increase awareness regarding the benefits deriving from shifting to sustainable or green modes of transport | behavioural change towards sustainable or green modes of transport | 4 | 11 | 2 | 4;5 | | |
| 19 | increase co-creation of new mobility solutions/ideas (achieving co-creation) | influencing decision making (achieving co-planning) | 2 | 4;5;6 | 3 | 7;8;9 | 1 | 3 |
| 20 | behavioural change towards sustainable or green modes of transport | | 4 | 11 | 2 | 4 | | |
| 21 | increase awareness regarding the benefits deriving from shifting to sustainable or green modes of transport | behavioural change towards sustainable or green modes of transport | 4 | 11 | | | | |
| 22 | behavioural change towards sustainable or green modes of transport | | 4 | 11 | 3 | 8;9 | | |
| 23 | increase awareness regarding the benefits deriving from shifting to sustainable or green modes of transport | | 4 | 11 | 3 | 8;9 | | |
| 25 | behavioural change towards sustainable or green modes of transport | | 4 | 11 | 2 | 4;5 | | |
| 26 | behavioural change towards sustainable or green modes of transport | Increase of walking and cycling trips | 4 | 11 | | | | |
| 27 | behavioural change towards sustainable or green modes of transport | | 4 | 11 | | | | |
| 30 | behavioural change towards sustainable or green modes of transport | increase awareness regarding the benefits deriving from shifting to sustainable or green modes of transport | 4 | 11 | | | | |

**Table A1.** *Cont.*

| Good Practice ID | MAIN OBJECTIVE | SECONDARY OBJECTIVE | SUMP Phase | SUMP Step | SUMP Phase | SUMP Step | SUMP Phase | SUMP Step |
|---|---|---|---|---|---|---|---|---|
| 31 | behavioural change towards sustainable or green modes of transport | | 4 | 11 | | | | |
| 32 | behavioural change towards sustainable or green modes of transport | | 4 | 11 | 2 | 5 | | |
| 35 | increase awareness regarding the benefits deriving from shifting to sustainable or green modes of transport | increase awareness regarding the importance of participating in planning/creation | 4 | 11 | 1 | 3 | 3 | 8;9 |
| 36 | behavioural change towards sustainable or green modes of transport | influencing decision making (achieving co-planning) | 4 | 11 | 2 | 4;5;6 | 3 | 7;8;9 |
| 37 | exchanging information with citizens and stakeholders | increase awareness regarding the benefits deriving from shifting to sustainable or green modes of transport | 4 | 11 | 2 | 4;5;6 | 1 | 3 |
| 38 | increase awareness regarding the benefits deriving from shifting to sustainable or green modes of transport | behavioural change towards sustainable or green modes of transport | 4 | 11 | | | | |
| 39 | increase awareness regarding the benefits deriving from shifting to sustainable or green modes of transport | | 4 | 11 | 1 | 1 | | |
| 41 | behavioural change towards sustainable or green modes of transport | | 4 | 10 | | | | |
| 42 | behavioural change towards sustainable or green modes of transport | | 4 | 10;11 | | | | |
| 43 | Increase of walking and cycling trips | Increase of population awareness | 4 | 11 | 3 | 8;9 | | |
| 44 | increase awareness regarding the benefits deriving from shifting to sustainable or green modes of transport | exchanging information with citizens and stakeholders | 4 | 12 | | | | |

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
