# Peer review of "Systems Thinking and Group Concept Mapping for Classification of Marketing Techniques in Mobility Plans"

_sustainability, doi:10.3390/su142416936_

Round 1

Reviewer 1 Report

I find the paper a good contribution to identifying the many issues to consider when dealing with marketing techniques in mobility plans. 

The section dedicated to Group Concept Mapping covers a variety of techniques. What I miss is identifying the different kinds of persons concerning their openness to new ideas and changes/innovations (innovators, early adopters, early majority, late majority, and laggards) and focusing on the different strategies to follow through time with each of these groups.

And the different possible stages: Awareness, Interest, Appraisal/Trial, and Adoption.

Also, I see the paper's focus as mainly on unidirectional communication. Bidirectional communication includes aspects the public considers relevant given their particular circumstances and times. Some digital Internet-based communication tools (in addition to the ones mentioned in the paper) may be helpful.

Another aspect that I wish to emphasize is that Systems Thinking is not equal (only) to System Dynamics. This association is common among many USA researchers.  

The field of Systems Thinking is a vast field that includes many approaches, methodologies, methods, technics, etc. It is common to mention several "waves" (each one with a specific perception of reality, complexity, and many other issues). For example (roughly speaking), the named first wave includes methodologies developed from the '50s to the 70s; the second wave includes approaches developed from the 70s to 90s, and the third wave extends from the 90s to the present. For example, in the cybernetic field (also systemic), we can deal with first-order cybernetics, second-order cybernetics, and third-order cybernetics. 

This comment does not affect my opinion about the quality of the paper at all. It only points out something that must be corrected because it is not true that ST=SD. ST>SD. The generalization that makes System Dynamics equivalent to Systems Thinking is not acceptable.

Reviewer 2 Report

I would like you to fix these errors in the text:

·         Too long title of the article. I highly suggest shortening the title of the article

·         Robert Magnuszewski missing * in line 5

·         154 -158 Great Footbal Allegory

·         201- 218 the Information duplicates like cluster analysis – “organizes and analyses it statistically using multidimensional scaling and cluster analysis. It involves a structured multi-step process [20] including data gathering, sorting and rating, multidimensional scaling and cluster analysis, and the generation and interpretation of multiple maps”

·         In 349-351, 353-355 and in more places the numbering is set in wrong place. Please fix it to a, b, c… or 1,2,3…

·         I had a problem with reading figure 1 information. I suggest to divide the figure 1 to two parts or change orientation of the page

·         Figure 2, 3 and 4 should be upscaled and resized to 120%.

·         Figure 5 the title “TOPSIS Results” inside figure is not necessary. Also the border

·         I suggest to add newer cited references

To sum it up A classification of marketing approaches and methods has been developed, providing numerous insights for application and understanding the dynamics underlying the transition to a more sustainable mobility state, beginning with a contextually dependent SUMP cycle phase. I appreciate the empirical results. 

Reviewer 3 Report

 The present paper proposes a hybrid method that can assist policy makers to categorize the marketing techniques towards the sustainable urban mobility plans steps, with the combination of Systems Thinking and modified Group Concept Mapping. I think the authors have written an interesting paper, dealing with an important topic. The overall representation of this paper is technically sound. I have, however, a few comments and suggestions for them:

1.      Abstract should be a little changed. To emphasize contributions and novelty.

2.      In the literature review, it is suggested to add the latest frontiers of literature published in 2021-22.

3.      The "Conclusions" section intends to help the reader understand why your research should matter to them after they have finished reading the paper. It is suggested to organize this section much better as it should be presented in one 250-300 words paragraph that contains unique results and findings. The description of future research directions should be extended in this Section.

Reviewer 4 Report

I would like to congratulate the authors on their work in preparing this article and on taking up the important subject of planning the development of sustainable mobility. Nevertheless, in my opinion, the article needs to be corrected, especially in terms of the construction and structure of the work.

1. The introductory chapter is too long and brings too much information – I suggest considering separating a section for the theoretical background, and in the introduction limit yourself to basic information about the purpose of the work and the importance of the subject matter.

2. Description of the purpose of the work seems a bit chaotic. Three goals of work have been set. Maybe it would be worth hierarchizing them? They have been proposed in such a way that one can also choose one of them as the goal of the work, and the others as research tasks subordinated to the assigned purpose of the work.

3. The methodological part also needs to be shortened. Descriptions of the next steps of the research procedure introduce too much information. Perhaps the solution would be to shorten the techniques described, or describe them in detail elsewhere. A graph showing the next steps would also be a good solution.

4. In the initial part of the Results chapter there should be no description of the method – in this chapter there should be only the results of the research carried out with a commentary. Due to the breadth of the resulting chapter, I would also suggest dividing it into subsections – to increase the clarity of the article.

5. There is no clearly separate discussion section in which the authors could comment on what the results of their work bring to the current state of knowledge and refer to other publications in the studied field.

6. The conclusions are formulated in a correct and very clear way.

Given the application significance of the SUMP topic raised in the article, I believe that the article should be published after the introduction of the major revision. SUMP is an increasingly used tool for shaping sustainable mobility, which is why any scientific study that evaluates the process of its creation and indicates the possibilities of improving this process is a valuable contribution to science. I hope that comments mainly on the structure and form of work will be introduced to make the article more clear.

Reviewer 5 Report

Article Review

November 15th, 2022

1. & 2. Relevance / Importance of Topic

The research paper “A hybrid Systems Thinking-Group Concept Mapping methodological framework for the classification of marketing techniques in mobility plans” is noteworthy topic for the industry. The purpose of the current paper is to offer a hybrid method that can assist policy makers to categorize the marketing techniques towards the sustainable urban mobility plans steps, with the combination of Systems Thinking and modified Group Concept Mapping. The paper concludes with a series of recommendations to policy makers on which marketing technique appears to be the most appropriate under different conditions.

3. Quality of analysis & evidence

Paper Strength

a)      The goal of the research paper “A hybrid Systems Thinking-Group Concept Mapping methodological framework for the classification of marketing techniques in mobility plans” is a relevant topic with a good positioning statement. However, statements in the paper needs to be structured and modified.

b)     Literature review – research paper extensively outlines available literature. However, it must be noted that there are some articles mentioned that are not recent. In this paper, it would have been great to do systematic literature analysis.

c)      Research analysisThe general objective of the proposed methodological framework is to classify the marketing techniques according to several relevant attributes and classes per step of the SUMP cycle. 8 step methodological flow was introduced which illustrated cohesive details. Moreover, in the paper there are great figures that show connection between variables.

d)     Conclusion - The conclusions are derived from the results. There is a good connection between results and research.

Paper Weakness

a)      Practical part of research – it is suggested that the author to present some tables that show the literature analysis.

b)     Grammatical issues – It is suggested that author cross-examine their writing and fix major bugs. There are heavy sentences that require simplification

c)      Research title: The research title is complicated. The authors are advised to reformat it.

d)     Marketing techniques: Moreover, in the title there is a mention about marketing techniques, however, in the paper there is no evidence what technique and how they are important for the solutions. Authors are advice to provide evidence on this issue.

e)       Tables: It is suggested that to move Tables 1, 2 and 3 into appendix, as there are not needed in the body of the paper.

4. Paper Description - Organization and presentation quality

The research paper " A hybrid Systems Thinking-Group Concept Mapping methodological framework for the classification of marketing techniques in mobility plans” is organized cohesively and follows an organized structure. Therefore, all the necessary parts are included in it. The introduction sets the tone of the research. The literature review illustrates valuable theoretical and practical information. Research methods and results are interconnected; however, it would have been suggested to have a different subsection about research methods and methodology.

To conclude, there are some formatting shortcomings.

5. Contribution to theory or practice

The research paper " A hybrid Systems Thinking-Group Concept Mapping methodological framework for the classification of marketing techniques in mobility plans” potentially contributes to a practical study, however, needs a major formatting.  

6. Overall quality

Overall, the authors have grasped a fascinating topic. The quality of the research paper " A hybrid Systems Thinking-Group Concept Mapping methodological framework for the classification of marketing techniques in mobility plans” is acceptable for the journal with major revision. Therefore, I recommend for Acceptance as full paper after major revision.

Round 2

Reviewer 4 Report

I would like to thank the authors for the corrections and their responses. The quality of the prepared article has been greatly improved. I believe that the article in its current form should be accepted for publication. Once again, congratulations to the Authors.

Author Response

We would like to thank the reviewer for their kind words. We would like also to state that we did an extensive editing of the paper to address any error in grammar and/or syntax.

Reviewer 5 Report

The authors did a good job applying suggestions. 

Author Response

we would like to thank the reviewer for their kind words. We have also checked the document again to correct any omissions and/or mistakes